# LEARNING TO PLAN
# WITH PERSONALIZED PREFERENCES

## ABSTRACT

Effective integration of Artificial Intelligence (AI) agents into daily life requires them to understand and adapt to individual human preferences, particularly in assistive roles. Although recent studies on embodied intelligence have advanced significantly, they typically adopt generalized approaches that overlook personalized preferences in planning. Cognitive research has demonstrated that these preferences serve as crucial intermediate representations in human decision-making processes and, though implicitly expressed through minimal demonstrations, can generalize across diverse planning scenarios. To systematically address this gap, we introduce the Preference-based Planning (PBP) benchmark, an embodied benchmark designed to evaluate agents' ability to learn preferences from few demonstrations and adapt their planning strategies accordingly. PBP features hundreds of diverse preferences spanning from atomic actions to complex sequences, enabling comprehensive assessment of preference learning capabilities. Evaluations of SOTA methods reveal that while symbol-based approaches show promise in scalability, significant challenges remain in learning to generate plans that satisfy personalized preferences. Building on these findings, we develop agents that not only learn preferences from few demonstrations but also adapt their planning strategies based on these preferences. Experiments in PBP demonstrate that incorporating learned preferences as intermediate representations significantly improves an agent's ability to construct personalized plans, establishing preference as a valuable abstraction layer for adaptive planning.

## 1 INTRODUCTION

The field of embodied Artificial Intelligence (AI) is rapidly advancing, driven by significant progress in foundation models for vision and language (Bommasani et al., 2021; Peng et al., 2023; Achiam et al., 2023; Bai et al., 2023). These advances enable AI systems to autonomously assist or collaborate with humans in daily tasks, particularly in domestic settings (Driess et al., 2023; Leal et al., 2023; Zitkovich et al., 2023; Ahn et al., 2024). However, recent approaches utilizing natural language instructions (Mu et al., 2023; Zitkovich et al., 2023; Singh et al., 2023) face fundamental limitations in capturing human preferences (Zhu et al., 2016). While natural language is our primary means of communication, its inherent ambiguity creates a gap between instructions and intended executions (Yuan et al., 2022; Jiang et al., 2022; 2021; Yuan et al., 2020). For instance, when a user requests help in preparing an apple, the agent needs to understand specific preferences about apple selection, washing requirements, cutting style, and container choice—details that vary significantly across individuals; see also Figure 1 for a graphical illustration.

Preference, central to personalization (Slovic, 1995), remains inadequately addressed in embodied Artificial Intelligence (AI). Integrating personalized preferences is crucial for tailoring agent actions to individual users, thereby enhancing the effectiveness and satisfaction of embodied assistants (Lee et al., 2012; Leyzberg et al., 2014). Moreover, preferences guide human-like decision-making and intelligent behavior. Psychological research emphasizes that understanding preferences is vital for interpreting human behaviors (Fawcett & Markson, 2010) and facilitating social interactions (Gerson et al., 2017; Liberman et al., 2021), suggesting that preference understanding could enable more grounded planning in embodied assistants.

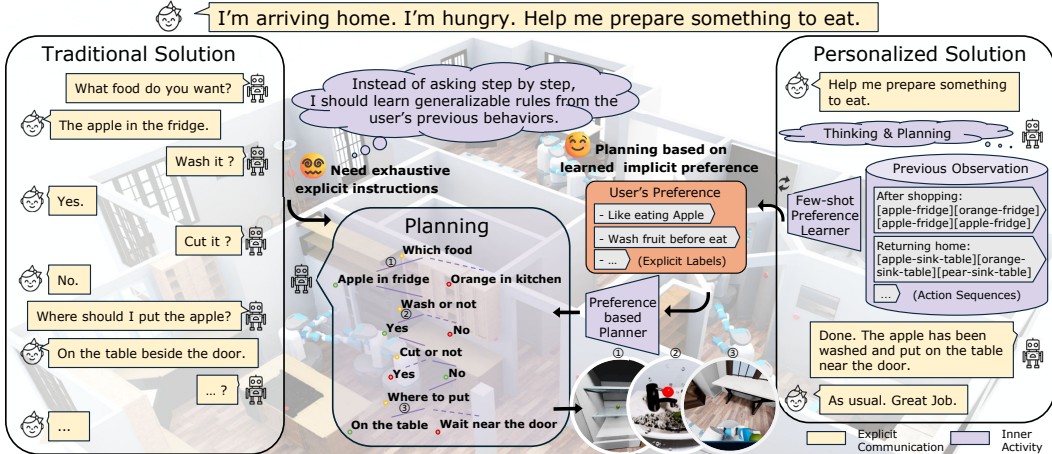

Figure 1: **An example of preference-based planning in a food preparation scenario.** When the assistant receives a natural language instruction for a food preparation task, it can follow one of two approaches: (Left, traditional methods) The assistant verifies details with the user at each step through exhaustive communication; or (Right, our personalized approach) it first learns from previous user action sequences to infer explicit preference labels and then generates a personalized plan based on the learned preferences. The planning tree (middle) illustrates how preferences guide the whole decision-making process across multiple dimensions. By learning preferences as a key intermediate representation from minimal human demonstrations, our approach enables AI agents to deliver personalized and adaptable assistance without explicit step-by-step instructions.

Learning human preferences in real-world settings presents unique challenges (Peng et al., 2024). Humans typically communicate their needs succinctly (Levinson, 1983), without exhaustive preference details (Lichtenstein & Slovic, 2006), and many preferences include unconscious or instinctive elements difficult to articulate (Epstein, 1994; Simonson, 2008). A practical approach is to infer preferences from observed human choices and decision-making patterns, as illustrated in Figure 1, where a robot assistant can learn users' preferences and behavior habits from previous observations.

In this paper, we focus on agents capable of learning preferences from human behavior and subsequently planning actions guided by these learned preferences. While previous studies have explored preference-based learning, they are limited to specific tasks or non-embodied scenarios (*e.g.*, rearrangement (Kapelyukh & Johns, 2022), scheduling (Yuan et al., 2023), dialogue generation (Ashby et al., 2023)) and fail to generalize across different situations. To address this limitation, we introduce Preference-based Planning (PBP), a comprehensive embodied benchmark built upon NVIDIA Omniverse and OmniGibson (Li et al., 2023a). PBP provides realistic simulation and real-time rendering for thousands of daily activities across 50 scenes, featuring a parameterized vocabulary of 290 diverse preferences. These preferences span multiple levels, from specific action-level preferences (*e.g.*, preferred glass type, water temperature) to task sequence-level preferences (*e.g.*, task ordering, subtask prioritization).

Given the expensive nature of data collection (Akgun et al., 2012) and the few-shot nature of preference acquisition, we frame preference learning as a few-shot learning from demonstration task. In this framework, agents must respond to ambiguous instructions by formulating plans aligned with preferences demonstrated in limited example sequences. Specifically, an agent needs to analyze behavioral data, identify consistent patterns, and extrapolate these patterns to higher-level preference abstractions that can generalize across various tasks (Chao et al., 2011). Furthermore, when confronted with new tasks, the agent should leverage these learned preferences to generate adaptive action sequences that align with user preferences while maintaining task efficiency.

With the PBP benchmark developed, we challenge existing learning agents on their ability to learn human preference and subsequently conduct preference-based planning. Our systematic evaluation of State-of-the-Art (SOTA) algorithms on PBP reveals that preferences serve as valuable abstractions of human behaviors, and their incorporation as intermediate planning steps significantly enhances agent adaptability. Through extensive experimentation, we demonstrate that symbol-based approaches show promise in scalability, yet significant challenges remain in both preference learning and planning. These challenges stem from the complexity of planning intricate activities and the nuanced nature of learning preferences through perception. Our analysis particularly highlights the

difficulties in few-shot preference learning and preference-guided planning, establishing preferences as a crucial abstraction layer between high-level goals and low-level actions. We present this work as a foundation for addressing these challenges in preference-based embodied AI.

## 2 RELATED WORK

### 2.1 THEORETICAL FOUNDATIONS OF HUMAN PREFERENCES

Preference theory originates from psychological research, where it describes predictable patterns in human behavior that can be modeled mathematically (Kahneman, 1982). These preferences reflect individual attitudes towards available choices in decision-making (Lichtenstein & Slovic, 2006) and operate both consciously and unconsciously to shape behavior (Coppin et al., 2010). A fundamental principle is that underlying preferences can be inferred from consistent behavioral patterns (Sen, 1973), enabling systematic analysis of decision-making processes. This framework has extended beyond psychology into economics, where Rational Choice Theory (Scott et al., 2000) models decision-making based on rational self-interest (Zey, 1998). Building on this, Utility Theory provides a mathematical foundation for modeling how preferences relate to attitudes toward rewards and risks (Mongin, 1997; Aleskerov et al., 2007). These theoretical foundations establish preferences as fundamental elements in shaping both individual behavior and broader societal dynamics. In recent years, these preference models have found new applications in artificial intelligence and robotics, particularly in developing human-centric AI assistants capable of understanding and adapting to individual user preferences.

### 2.2 EMBODIED TASK PLANNING BENCHMARKS

The development of intelligent embodied planning has evolved from basic Vision-and-Language Navigation (VLN) tasks (Anderson et al., 2018; Chen et al., 2019; Thomason et al., 2020) to complex interactive scenarios. ALFRED (Shridhar et al., 2020) introduced object manipulation, state tracking, and temporal dependencies between instructions, and platforms like Habitat (Savva et al., 2019; Puig et al., 2023b) and AI2-THOR (Kolve et al., 2017) emphasize active perception, long-term planning, and interactive learning, while VisualAgentBench (Liu et al., 2024b) and Embodied-Bench Yang et al. (2025) focus on abilities of visual foundation agents. Recent research has also shifted toward implicit-instruction scenarios particularly in arrangement (Taniguchi et al., 2021; Kant et al., 2022; Sarch et al., 2022), where agents reason about object placements without explicit directives. These works leverage commonsense knowledge or **general preferences** to execute universally-accepted behavioral norms, but neglect nuanced variations among different people.

Our work emphasizes **personalized preferences**, where embodied agents align their actions with individual user habits. This includes personalized object placement strategies (Abdo et al., 2015; Kapelyukh & Johns, 2022; Wu et al., 2023) in housekeeping, multi-agent coordination where agents adapt to individual preferences to achieve optimal coordination (Shu & Tian, 2019; Puig et al., 2021b; 2023a). Works on proactive assistance (Patel & Chernova, 2023; Patel et al., 2023) further explore anticipating temporal patterns in humans' daily routines. We extend these chanllenges by considering preferences across diverse situations and scenes. Beyond object placements, we address temporal action sequences, state transitions during interactions, and few-shot preference learning, thus further enabling robust preference modeling and adaptation in real-world scenarios.

### 2.3 PERSONALIZATION IN FOUNDATION MODELS

The problem of adapting foundation models to distinct users has raised great attention (Tseng et al., 2024). Existing studies explore prompting methods or Reinforcement Learning from Human Feedback (RLHF) to align models to user preferences in recommendation (Yang et al., 2023; Wang & Lim, 2023), education (Huber et al., 2024; Park et al., 2024), healthcare (Zhang et al., 2024b), and dialogue generation (Li et al., 2023b; Ashby et al., 2023; Song et al., 2020; Lee et al., 2024). However, while these foundation models excel at learning and adapting to individual preferences in conversational settings, their ability in the situation of limited demonstrations and embodied planning remains an open challenge, particularly in multi-step tasks requiring complex perceptual understanding and subtle execution strategies. Utilizing Large Language Models (LLMs) as few-shot planners to generate language-based action sequences from limited demonstrations (Song et al., 2023; Driess et al.,

2023; Ding et al., 2023; Zhang et al., 2024a), or using Vision-Language Models (VLMs) to enhance robotic systems' perception and reasoning capabilities (Ahn et al., 2024; Leal et al., 2023; Gu et al., 2023; Brohan et al., 2022; Zitkovich et al., 2023; Xu et al., 2024a) has proven to be effective. Yet, how to incorporate personalization into them for embodied planning is not fully addressed.

## 3 FORMULATING PREFERENCE-BASED PLANNING

Tasks in PBP mirror real-world watch-and-help scenarios (Puig et al., 2021a), where an agent observes a few demonstrations of a user performing tasks that reveal preferences. The agent must then complete similar tasks in different setups while adhering to the demonstrated preferences.

Preference-based planning comprises two key components: few-shot **preference learning** of user preferences and subsequent **planning** guided by these learned preferences. Since humans, even infants, can naturally detect others' preferences from limited decisions (Choi & Luo, 2023), and collecting extensive personal demonstrations is impractical in daily life, we formulate this as few-shot learning from demonstration.

Given a user with preference $\mathbf{p}$, the agent observes the user performing tasks from a first-person perspective, denoted as $\mathbf{O}$. These observations span multiple demonstrations. Formally, $\mathbf{O}$ contains both state and action observations: $\mathbf{O} = \{(\mathcal{S}_i, \mathcal{A}_i, \mathcal{M})_N\}$, where $\mathcal{S}_i$ denotes the egocentric observation sequence in the $i$-th demonstration, $\mathcal{A}_i$ represents the action sequence, and $\mathcal{M}$ optionally provides a bird's-eye view of the entire scene map.

In the first stage, the objective is to learn the preference representation demonstrated through user actions:

$$\mathbf{p} = f(\mathbf{O}; \theta_f), \tag{1}$$

where $\mathbf{p}$ denotes the learned preference representation here. It can either be a hidden representation or an explicit textual label, depending on the task settings.

The learned preference $\mathbf{p}$ should then guide planning when the agent faces different setups with varying objects, room layouts, or entire scenes. Specifically, the agent optimizes:

$$\mathcal{L} = \sum_{i=1} \ell(g(s_i, f(\mathbf{O}; \theta_f); \theta_g), a_i), \tag{2}$$

where $g(\cdot)$ represents a potentially parameterized planning function that maps the current state and preference representation to the next action, and $a_i$ denotes the ground-truth action demonstrating the user's preference at the current stage.

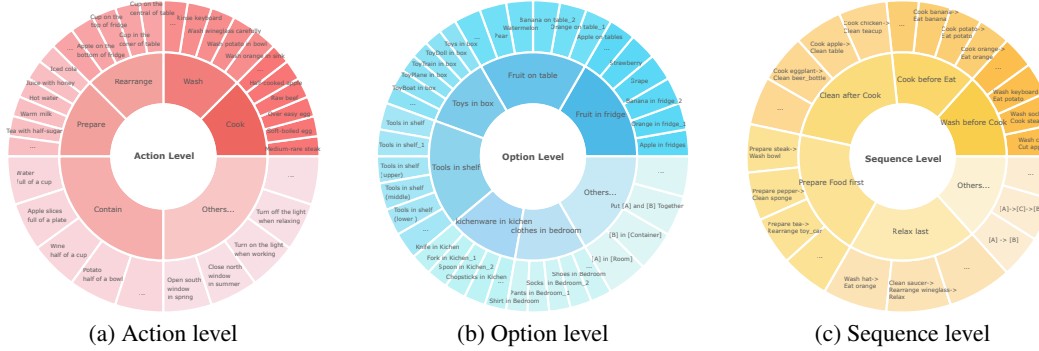

|  |  |  |
|---|---|---|
| (a) Action level | (b) Option level | (c) Sequence level |

Figure 2: **Hierarchical organization of user preferences.** Our framework organizes preferences in a three-tiered structure, visualized through sunburst diagrams: (a) Action level captures fine-grained execution details within specific tasks, from quantity preferences in "Contain" (*e.g.*, "half a cup" *vs.* "full cup") to environmental controls (*e.g.*, lighting and window operations). (b) Option level represents spatial preferences for object categories, encoding both storage decisions (*e.g.*, table *vs.* fridge for fruits) and organizational choices (*e.g.*, shelf levels and boxes for tools/toys). (c) Sequence level defines temporal relationships between tasks, encompassing both basic preparation sequences (*e.g.*, "Prepare Food first") and conditional orderings (*e.g.*, "Clean after Cook," "[A]->[B]"). Each diagram's hierarchical structure branches from general categories to specific instances, revealing detailed preference patterns upon closer inspection. (Vector graphics; zoom in for details.)

# 4 THE PREFERENCE-BASED PLANNING (PBP) BENCHMARK

Built on NVIDIA's Omniverse and OmniGibson simulation environment (Li et al., 2023a), our PBP benchmark enables realistic simulation of thousands of daily activities. It spans 50 distinct scenes and encodes 290 unique preferences, with a comprehensive test set of 5000 instances. Below, we detail the preference structure and test set construction.

## 4.1 DEFINITION OF PREFERENCES

We organize preferences in a three-tiered hierarchical structure that captures varying degrees of specificity across tasks. Figure 2 provides an overview of all preferences and their distribution, while Figure 3 illustrates concrete examples of preferences and corresponding agent actions. The 290 preferences are distributed across three levels: 80 for sequence-level, 135 for option-level, and 75 for action-level preferences.

**Action Level** These bottom-level preferences govern fine-grained execution details within specific sub-tasks, such as water quantity preferences when filling cups or shelf placement choices for books.

**Option Level** Middle-level preferences encode alternative approaches to sub-tasks. For instance, in "storing-nonperishable-food," users may prefer cabinet storage versus table placement. These preferences can bind to different objects and may compose multiple action-level preferences.

**Sequence Level** Top-level preferences define task ordering and prioritization. They capture temporal dependencies between sub-tasks, such as cleaning furniture before rearranging kitchen utensils, followed by dinner preparation upon returning home.

## 4.2 CONSTRUCTING PBP TEST SET

Our PBP benchmark includes a default test set for systematic evaluation. Following the formulation in Section 3, we structure PBP tasks as few-shot learning-from-demonstration problems. Each test point comprises several (typically three) unique demonstrations with egocentric observations of action sequences and their corresponding preference labels. As illustrated in Figure 4, a demonstration includes an egocentric video of agent activity, a bird's-eye-view map tracking agent position, and frame-level action annotations. We also provide third-person view recordings for enhanced visualization. We prioritize the egocentric perspective for two reasons: 1) it offers a clear view with minimal occlusions, and 2) it aligns with human perception, facilitating transfer to real-world data from head-mounted devices.

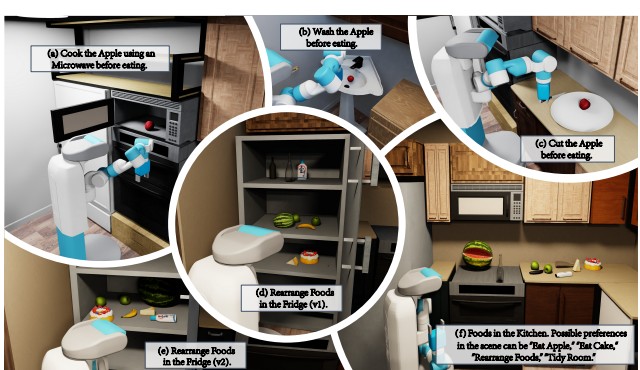

Figure 3: **Example of preferences and their corresponding actions in PBP.** Demonstration of preference hierarchies in PBP: (a-c) Primitive actions (cooking, washing, cutting); (d-e) Object rearrangement strategies (category grouping vs. fridge-layer placement); (f) Task sequencing (fruits before cleaning). (Vector graphics; zoom in for details.)

The test set construction follows a two-stage process. First, we build a reusable and extensible demonstration pool. To generate each demonstration, we randomly assign a preference from our defined primitives to one of 50 OmniGibson scenes, then sample relevant objects within the chosen scene. We generate multi-perspective observations using rule-based planners for high-level planning and predefined scripts for low-level execution (*e.g.*, Inverse Kinematics (IK) for grasping, A* for movement).

Second, we construct test points by sampling preferences and retrieving relevant demonstrations from the pool. To reflect real-world few-shot scenarios, each preference is paired with demonstrations that share the same high-level preference but vary in scene settings or object selections. We also include unrelated demonstrations to prevent sampling bias.

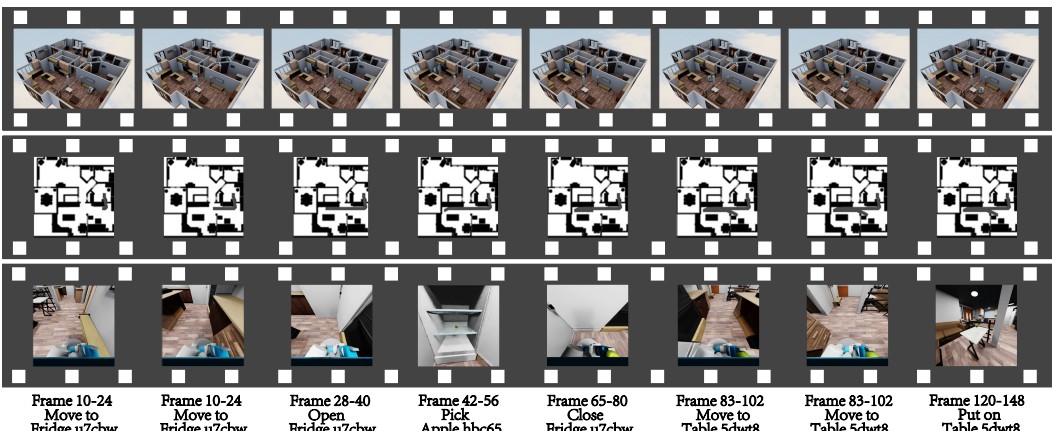

Frame 10-24 Move to Fridge u7cbw    Frame 10-24 Move to Fridge u7cbw    Frame 28-40 Open Fridge u7cbw    Frame 42-56 Pick Apple hbc65    Frame 65-80 Close Fridge u7cbw    Frame 83-102 Move to Table 5dwt8    Frame 83-102 Move to Table 5dwt8    Frame 120-148 Put on Table 5dwt8

Figure 4: **Example of a demonstration in PBP.** The robot in the demonstration is executing the task "Pick Apple from Fridge and place on Table". **Top:** A third-person view video provides an overhead perspective of the entire scene. **Middle:** The bird's-eye-view map displays the robot's relative position within the scene. **Bottom:** The egocentric video captures the robot's first-person observations during task execution. **Text:** The per-frame action annotations contain Omniverse object IDs, which ensure each object reference is unique and enable the model to identify specific objects precisely.

The default test set contains 5,000 test points, drawing from a pool of 15,000 unique recordings. Unless specified otherwise, all experiments use this default set. The benchmark also supports custom test point generation through flexible demonstration sampling, preference definition, and third-person view video creation.

### 4.3 MODELS

Our evaluation focuses primarily on multimodal models that incorporate LLMs and demonstrate strong few-shot learning capabilities. The LLM component serves as a knowledge base that can enhance preference learning through commonsense reasoning. We also include symbol-based LLM models for ablation studies to analyze how different modalities impact PBP performance. Most models evaluated can function in both end-to-end and two-stage pipeline configurations. See Appendix D for detailed implementations.

## 5 EXPERIMENTS

### 5.1 EXPERIMENTAL SETUP

We evaluate preference learning capabilities across two distinct settings: end-to-end and two-stage approaches. In the end-to-end setting, models directly map raw state inputs to action outputs. Leveraging models' in-context learning abilities, we provide demonstrations alongside current state information as input and evaluate the generated action sequences against ground truth.

The two-stage setting introduces an intermediate step where models first learn to predict explicit preference labels during training. These predicted labels then serve as preference representations for subsequent planning stages. For black-box models, we employ carefully designed prompts rather than fine-tuning approaches.

All demonstration videos maintain consistent technical specifications across models and agents: egocentric perspective, $512 \times 512$ resolution, and 8 fps frame rate. Video duration matches the corresponding action sequence length. For LLM inference, we use conservative decoding parameters: temperature of 0.05, top-k of 1, and top-p of 0.05. All experiments run on a single machine with 8 NVIDIA A100 GPUs.

### 5.2 END-TO-END ACTION PREFERENCE LEARNING

We first evaluate model performance in the end-to-end setting, where models generate actions directly from previous demonstrations and current state information. To quantify performance, we

Table 1: **Levenshtein distance between generated and ground truth action sequences. End-to-end** represents models directly generating action sequences from demonstration-preference pair examples. **Two-stage** indicates generation using both demonstrations and previously inferred preference labels based on demonstrations. **Second-stage (gt)** uses demonstrations alongside ground truth preference labels for sequence generation.

| | VIDEO-BASED INPUT | | | | SYMBOL-BASED INPUT | | |
|---|---|---|---|---|---|---|---|
| | ViViT | LLaVA-Next | EILEV | GPT-4V | Llama3-8B | DeepSeek-R1 | GPT-4.1 |
| **Option Level** | | | | | | | |
| **End-to-end** | 15.49±1.29 | 15.94±3.41 | 12.88±2.20 | 15.63±2.31 | 14.74±3.21 | 8.73 ±3.03 | 7.42±2.67 |
| **Two-stage** | - | 12.46±3.23 | 12.89±3.74 | 8.37±2.19 | 9.67±5.16 | 3.19±2.19 | 2.26±2.03 |
| **Second-stage (gt)** | - | 3.28±5.29 | 11.18±4.20 | 1.26±2.55 | 8.22±5.58 | 1.76±1.89 | 0.15±2.85 |
| **Sequence Level** | | | | | | | |
| **End-to-end** | 34.04±11.84 | 34.76±11.25 | 33.10±12.21 | 33.75±11.15 | 31.79±7.32 | 28.72±4.12 | 26.48±3.25 |
| **Two-stage** | - | 30.02±13.54 | 33.03±13.61 | 27.52±9.48 | 25.46±5.93 | 18.61±3.25 | 14.19±3.01 |
| **Second-stage (gt)** | - | 18.92±14.18 | 26.57±12.21 | 11.36±8.05 | 19.02±7.10 | 14.10±3.76 | 10.31±2.98 |
| **Overall** | | | | | | | |
| **End-to-end** | 24.76 | 25.35 | **22.99** | 24.69 | 23.26 | 18.72 | **16.95** |
| **Two-stage** | - | 21.24 | 22.96 | **17.94** | 17.56 | 10.90 | **8.22** |
| **Second-stage (gt)** | - | 11.10 | 18.88 | **6.31** | 13.62 | 7.93 | **5.23** |

use Levenshtein distance to measure discrepancies between generated and ground truth action sequences, treating each individual action as a token.

As shown in Table 1 (the `End-to-end` row), video-based models produce Levenshtein distances approaching the average ground truth sequence lengths (15.80 at option level, 35.87 at sequence level). These high distances indicate that the models generate predominantly inconsistent action sequences, suggesting a failure to grasp preferences embedded in demonstration videos. While symbol-based models show modest improvements, their performance gains remain limited.

These findings expose a fundamental limitation in current models: they struggle to extract underlying relationships from perceptual inputs without explicit intermediate guidance. The models appear to learn individual, isolated actions rather than cohesive action patterns that reflect implicit preferences. This significant gap underscores the inherent challenge of performing end-to-end preference learning solely from demonstrations.

### 5.3 TWO-STAGE LEARNING-PLANNING

Given the limitations of end-to-end learning, we implement a two-stage approach to decompose the preference learning problem. The first stage focuses on preference prediction, where we provide models with auxiliary preference token labels and train them to predict hidden preferences explicitly. These preference tokens, as discussed in Section 4.1, maintain sufficient semantic content for translation into primitive actions.

Results from the first stage (Table 2) reveal significant performance variations across models. At the option level, GPT-4V achieves superior performance with 48.48% accuracy, demonstrating strong capability in interpreting demonstrated preferences. Among symbol-based models, the stark contrast between DAG-Opt's limited performance and the improved results from Llama3-8B and GPT-4.1 highlights the advantage of next-token prediction over dependency learning for preference inference. Models with language components consistently show improved preference understanding compared to end-to-end learning.

Table 2: **Preference prediction accuracy in few-shot and ablative settings.**

| | VIDEO-BASED INPUT | | | | SYMBOL-BASED INPUT | | | |
|---|---|---|---|---|---|---|---|---|
| | ViViT | LLaVA-Next | EILEV | GPT-4V | DAG-Opt | Llama3-8B | DeepSeek-R1 | GPT-4.1 |
| **Few-shot** | | | | | | | | |
| **Option Level** | 9.38 | 36.87 | 38.33 | 48.48 | 10.15 | 72.98 | 86.02 | 88.91 |
| **Sequence Level** | 4.24 | 24.85 | 32.69 | 37.50 | 13.49 | 67.18 | 71.21 | 70.28 |
| **Overall** | 6.81 | 30.86 | 35.51 | **42.99** | 11.82 | 70.08 | 78.62 | **79.60** |
| **Ablative** | | | | | | | | |
| **Option Level** | 9.16 | 15.47 | 4.77 | 29.42 | 3.84 | 39.50 | 78.19 | 75.29 |
| **Sequence Level** | 4.38 | 8.13 | 0.00 | 0.00 | 1.28 | 6.25 | 15.29 | 14.11 |
| **Overall** | 6.77 | 11.8 | 2.38 | **14.71** | 2.56 | 22.88 | **46.74** | 44.70 |

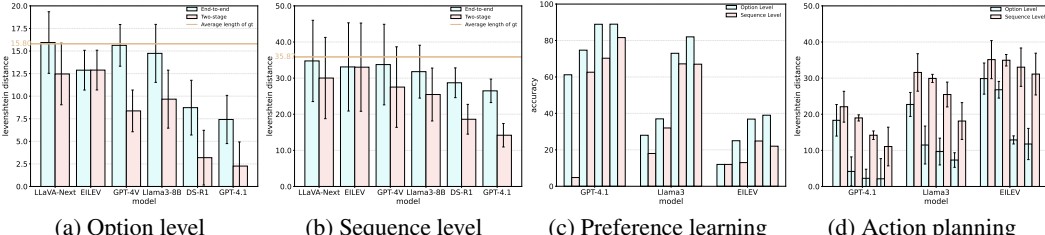

(a) Option level      (b) Sequence level      (c) Preference learning      (d) Action planning

Figure 5: **(a)-(b) Levenshtein distance between generated and ground truth action sequences.** Results shown for both (a) option level and (b) sequence level under two conditions: End-to-end bars represent direct sequence generation from previous observations, while Two-stage bars show performance when models receive predicted preference labels. The — line indicates average ground truth sequence length (option level: 15.80, sequence level: 35.87). Results demonstrate significantly improved performance under the two-stage approach compared to end-to-end generation. **(c)-(d) Ablation study on the number of demonstrations.** Models are evaluated across both of the two stages: (c) first-stage preference learning and (d) second-stage action planning. We evaluate both Option Level and Sequence Level tasks. The number of few-shot demonstrations varies from [1, 2, 3, 5], presented left to right for (c), higher accuracy indicates better performance. For (d), lower distance indicates better performance. Increased demonstration quantity generally improves both preference learning capability and planning effectiveness.

The second stage involves generating action sequences based on both demonstrations and predicted preference labels from the first stage, introducing potential error propagation. Results in Table 1 ( Two-stage row) and Figure 5 (a)-(b) show significant improvements when models receive explicit preferences. For comprehensive evaluation, we include planning results using ground truth preference labels ( Second-stage (gt) row). GPT-4V and GPT-4.1 achieve near-zero Levenshtein distances, indicating almost perfect alignment with ground truth action sequences.

Analysis of both stages reveals distinct challenges across model types. Vision-based models like LLaVA-Next and GPT-4V struggle with preference inference but excel in action planning given preference labels, suggesting difficulty in abstracting preferences from visual input. Symbol-based models perform well in both preference inference and preference-guided planning, yet underperform in end-to-end settings. This indicates that models may lack innate preference-based reasoning capabilities but can effectively plan when preferences are explicitly provided.

To isolate the impact of prior knowledge versus in-context learning, we conduct ablation studies by removing demonstrations and testing preference prediction on isolated test sequences. Results in Table 2 (bottom) show significant performance degradation compared to few-shot learning (Table 2 (top)), particularly at the sequence level. This suggests that while models may encode basic task-specific preferences, they rely heavily on demonstrations to recognize complex preference patterns in varied sequences.

Table 3: **Models' generalization ability.** *direct* denotes experiments *without* generalization. *gen* denotes experiments conducted *with* generalization cases. We report the preference prediction accuracy.

| | LLaVA-Next | EILEV | GPT-4V | DeepSeek-R1 | GPT-4.1 |
|---|---|---|---|---|---|
| **Option Level** *direct* | 33.25 | 46.93 | 53.24 | 86.02 | 88.91 |
| **Option Level** *gen* | 36.87 | 38.33 | 48.48 | 84.98 | 87.12 |
| **Sequence Level** *direct* | 33.12 | 37.53 | 39.42 | 71.21 | 70.28 |
| **Sequence Level** *gen* | 24.85 | 32.69 | 37.50 | 70.16 | 68.01 |

## 5.4 GENERALIZATION

While human actions may vary across different objects and scenes, underlying preferences often remain consistent. We evaluate the models' ability to generalize preference learning across varying visual contexts. The original test set inherently tests generalization by randomly sampling scenes and objects when rendering video demonstrations for each preference. To gain additional insights, we conduct complementary experiments with controlled conditions where demonstration and test videos are rendered in identical rooms with the same objects. This controlled setting enables direct performance comparisons under consistent conditions. We evaluate Efficient In-context Learning on Egocentric Videos (EILEV), Large Language and Vision Assistant (LLaVA), and GPT-4.1 series

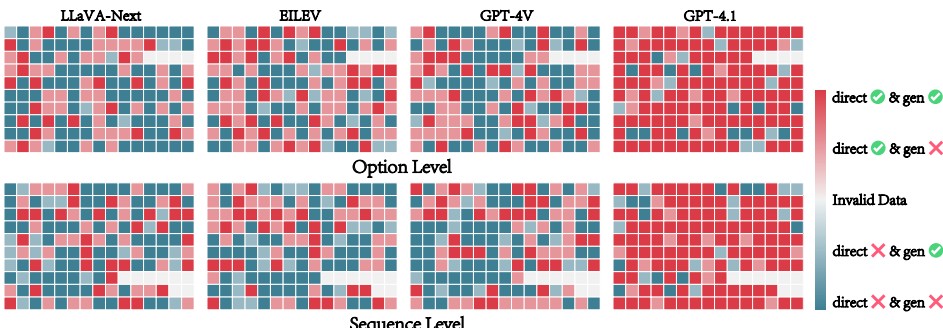

Figure 6: **Analysis of test samples in *direct* and *generalization* settings.** Lines represent distinct scenes, with grid colors indicating different sample statuses (*direct* for no generalization and *gen* for the oppsite).

models on this variant of PBP, as these models previously demonstrated strong few-shot reasoning capabilities. Results are summarized in Table 3.

Symbol-based reasoning models demonstrate consistent performance regardless of scene or object variations, while vision-based models show greater sensitivity to scene changes. This distinction stems from the nature of our predefined preferences, which are sufficiently abstract and general to apply across diverse scenes and objects. Vision-based models, however, tend to anchor their few-shot learned preferences to specific visual features of scenes or objects. When these visual elements change, preference recognition accuracy may deteriorate. This contextual dependence remains a persistent challenge for vision-based models, which often overfit to scene-specific features from training videos.

Analysis of test points across *direct* and *gen* conditions (Figure 6) reveals two key findings: (i) Preference learning performance correlates with scene characteristics, with certain scenes proving consistently challenging across both conditions. (ii) While *direct* cases show better performance overall, failure patterns differ between conditions, particularly for vision-based models. This suggests models rely heavily on visual context consistency–including object arrangement and scene layout–for accurate predictions, indicating potential superficial learning rather than true preference understanding. Symbol-based reasoning maintains robust performance across varied scenes due to the general nature of predefined preferences, whereas vision-based models' strong dependence on specific visual contexts limits their generalization capability.

## 5.5 ABLATIONS ON DEMONSTRATION NUMBERS

We examine the effect of demonstration quantity on model performance through an ablation study (Figure 5 (c)-(d)). Results show that increasing demonstration numbers generally improves preference learning and planning effectiveness. This improvement is most evident in second-stage planning, where models achieve lower sequence distances by more accurately replicating human actions. Models like GPT-4.1, Llama3, and EILEV show consistent performance gains with additional demonstrations. However, we observe that excessive demonstrations (*e.g.*, 5-demo cases for GPT-4.1 and EILVE) can sometimes impair first-stage prediction accuracy. Despite these occasional exceptions, the overall trend confirms our intuition: more demonstrations enhance learning and planning performance. These findings highlight the importance of demonstration quantity in developing effective personalized planning systems that align with user preferences.

## 6 CONCLUSION

We investigate embodied agents that learn and implement human preferences through behavioral observation and user interaction. We present Preference-based Planning (PBP), a comprehensive embodied benchmark designed to capture the complexity of real-world human preferences. An evaluation framework is developed to assess models' preference learning and implementation capabilities. Our findings demonstrate that preferences effectively abstract human behaviors and guide planning processes. While current models still face challenges in preference inference and adaptive planning from limited observations, incorporating preference-based reasoning improves both effectiveness and generalization. We aim to stimulate further research in this crucial yet understudied domain of developing preference-aware embodied agents.

ETHICS STATEMENT

The development of personalized, preference-based embodied AI agents has the potential to significantly enhance human-AI collaboration, improving efficiency and user satisfaction in a variety of daily tasks. As AI agents increasingly adapt to individual behaviors, ensuring that these systems respect user autonomy and data privacy is crucial. Given our focus on private scenarios, we anticipate minimal negative societal impact from this research.

REPRODUCIBILITY STATEMENT

We have provided source code and running tutorials as supplementary materials. Key implementation details and experimental settings are described in the main paper (Section 5 and Appendix D). We will open-source both the code and the dataset used in our experiments upon acceptance.

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

# A  DATASET CARD

We follow the datasheet proposed in Gebru et al. (2021) for documenting our proposed PBP:

1. **Motivation**

    (a) **For what purpose was the dataset created?**
    The benchmark was created to evaluate existing learning agents on their ability to understand and adapt to various human preferences. Specifically, it aims to test the agents' proficiency in few-shot learning from demonstrations, where they must respond to ambiguous task instructions and formulate adaptive task plans based on limited examples of user preferences. The benchmark is designed to highlight the challenges and gaps in current AI systems' capabilities in planning activities and abstracting human preferences, ultimately driving advancements towards developing more intelligent and personalized embodied agents.

    (b) **Who created the dataset and on behalf of which entity?**
    N/A.

    (c) **Who funded the creation of the dataset?**
    N/A.

    (d) **Any other Comments?**
    None.

2. **Composition**

    (a) **What do the instances that comprise the dataset represent?**
    Each instance contains an egocentric video of an agent's activity, its bird's-eye-view map of the position of the agent, and a frame-level textual annotation of the current action, as shown in Figure 4. Additionally, we provide a rendered third-person view of the entire process.

    (b) **How many instances are there in total?**
    15000.

    (c) **Does the dataset contain all possible instances or is it a sample (not necessarily random) of instances from a larger set?**
    No. The dataset contains a set of demonstrations rendered within the simulator, The users can render more diverse instances if they want. We have provided the rendering instructions.

    (d) **What data does each instance consist of?**
    The instances that comprise the benchmark represent various types of human preferences applied to different tasks within a realistic embodied scene. Each instance is designed to challenge the learning agents to understand and adapt to these preferences based on a few demonstration examples, reflecting the diverse and hierarchical nature of user preferences in real-world scenarios. See above for data details.

    (e) **Is there a label or target associated with each instance?**
    Yes.

    (f) **Is any information missing from individual instances?**
    No.

    (g) **Are relationships between individual instances made explicit?**
    Yes.

    (h) **Are there recommended data splits?**
    No.

    (i) **Are there any errors, sources of noise, or redundancies in the dataset?**
    No.

    (j) **Is the dataset self-contained, or does it link to or otherwise rely on external resources (*e.g.*, websites, tweets, other datasets)?**
    Self-contained.

    (k) **Does the dataset contain data that might be considered confidential (*e.g.*, data that is protected by legal privilege or by doctor-patient confidentiality, data that includes the content of individuals' non-public communications)?**
    No.

    (l) **Does the dataset contain data that, if viewed directly, might be offensive, insulting, threatening, or might otherwise cause anxiety?**
    No.

    (m) **Does the dataset relate to people?**
    No.

    (n) **Does the dataset identify any subpopulations (*e.g.*, by age, gender)?**
    No.

    (o) **Is it possible to identify individuals (*i.e.*, one or more natural persons), either directly or indirectly (*i.e.*, in combination with other data) from the dataset?**
    No.

(p) **Does the dataset contain data that might be considered sensitive in any way (*e.g.*, data that reveals racial or ethnic origins, sexual orientations, religious beliefs, political opinions or union memberships, or locations; financial or health data; biometric or genetic data; forms of government identification, such as social security numbers; criminal history)?**
No.

(q) **Any other comments?**
None.

3. **Collection Process**

(a) **How was the data associated with each instance acquired?**
We render PBP using NVIDIA's Omniverse and OmniGibson simulation environment (Li et al., 2023a).

(b) **What mechanisms or procedures were used to collect the data (*e.g.*, hardware apparatus or sensor, manual human curation, software program, software API)?**
The data for each instance in the benchmark was acquired by sampling preferences from a predefined set and constructing tasks paired with a few demonstrations that shared high-level preferences but differed in specific objects and scenes. Each sampled preference was randomly assigned to one of the 50 scenes provided by OmniGibson, with relevant objects sampled within the scene. Egocentric observation and action sequences of an embodied agent were generated as the agent performed tasks guided by a rule-based planner using planning primitives like inverse kinematics for grasping and the A* algorithm for movement.

(c) **If the dataset is a sample from a larger set, what was the sampling strategy (*e.g.*, deterministic, probabilistic with specific sampling probabilities)?**
N/A.

(d) **Who was involved in the data collection process (*e.g.*, students, crowdworkers, contractors) and how were they compensated (*e.g.*, how much were crowdworkers paid)?**
N/A.

(e) **Over what timeframe was the data collected?**
N/A.

(f) **Were any ethical review processes conducted (*e.g.*, by an institutional review board)?**
The dataset raises no ethical concerns.

(g) **Does the dataset relate to people?**
No.

(h) **Did you collect the data from the individuals in question directly, or obtain it via third parties or other sources (*e.g.*, websites)?**
N/A.

(i) **Were the individuals in question notified about the data collection?**
N/A.

(j) **Did the individuals in question consent to the collection and use of their data?**
N/A.

(k) **If consent was obtained, were the consenting individuals provided with a mechanism to revoke their consent in the future or for certain uses?**
N/A.

(l) **Has an analysis of the potential impact of the dataset and its use on data subjects (*e.g.*, a data protection impact analysis) been conducted?**
Yes.

(m) **Any other comments?**
None.

4. **Preprocessing, Cleaning and Labeling**

(a) **Was any preprocessing/cleaning/labeling of the data done (*e.g.*, discretization or bucketing, tokenization, part-of-speech tagging, SIFT feature extraction, removal of instances, processing of missing values)?**
N/A.

(b) **Was the "raw" data saved in addition to the preprocessed/cleaned/labeled data (*e.g.*, to support unanticipated future uses)?**
N/A.

(c) **Is the software used to preprocess/clean/label the instances available?**
N/A.

(d) **Any other comments?**
None.

5. **Uses**

(a) **Has the dataset been used for any tasks already?**
No, the dataset is newly proposed by us.

(b) **Is there a repository that links to any or all papers or systems that use the dataset?**
No, the dataset is new.

(c) **What (other) tasks could the dataset be used for?**
This dataset could be used for research topics like embodied AI and human-computer interaction.

(d) **Is there anything about the composition of the dataset or the way it was collected and preprocessed/cleaned/labeled that might impact future uses?**
N/A.

(e) **Are there tasks for which the dataset should not be used?**
N/A.

(f) **Any other comments?**
None.

6. **Distribution**

(a) **Will the dataset be distributed to third parties outside of the entity (*e.g.*, company, institution, organization) on behalf of which the dataset was created?**
No before it is made public.

(b) **How will the dataset be distributed (*e.g.*, tarball on website, API, GitHub)?**
On our project website upon acceptance.

(c) **When will the dataset be distributed?**
Upon acceptance.

(d) **Will the dataset be distributed under a copyright or other intellectual property (IP) license, and/or under applicable terms of use (ToU)?**
Under CC BY-NC [1] license.

(e) **Have any third parties imposed IP-based or other restrictions on the data associated with the instances?**
No.

(f) **Do any export controls or other regulatory restrictions apply to the dataset or to individual instances?**
No.

(g) **Any other comments?**
None.

7. **Maintenance**

(a) **Who is supporting/hosting/maintaining the dataset?**
The authors.

(b) **How can the owner/curator/manager of the dataset be contacted (*e.g.*, email address)?**
N/A.

(c) **Is there an erratum?**
Future erratum will be released through the website.

(d) **Will the dataset be updated (*e.g.*, to correct labeling errors, add new instances, delete instances')?**
Yes.

(e) **If the dataset relates to people, are there applicable limits on the retention of the data associated with the instances (*e.g.*, were individuals in question told that their data would be retained for a fixed period of time and then deleted)?**
N/A. The dataset does not relate to people.

(f) **Will older versions of the dataset continue to be supported/hosted/maintained?**
Yes.

(g) **If others want to extend/augment/build on/contribute to the dataset, is there a mechanism for them to do so?**
Yes. We will release the source code as well as a licence on our project website after acceptance.

(h) **Any other comments?**
None.

## B  DATASET STATISTICS

The length of the simulations in the dataset ranges from 1 to 5 minutes, depending on the tasks recorded. And the videos are recorded at 30 fps.

---

[1] https://creativecommons.org/licenses/by-nc/4.0/

## B.1 PREFERENES

See Table A1 for the preference statistics in PBP.

Table A1: **Dataset Statistics in PBP.**

|  | **Action_Level** | **Option_Level** | **Sequence_Level** |
|---|---|---|---|
| **Preference Num** | 75 | 135 | 80 |
| **Video Num** | 5000 | 5000 | 5000 |
| **Sub-task Num** | 1 | 2-3 | 2-3 |

## B.2 ACTIONS

See Table A2 for the action statistics in PBP. We implement 17 action primitives in PBP to assist with model planning and dataset rendering. These action primitives have parameters that simplify tasks and are considered the lowest-level actions. Each sub-task contains 8 to 20 such lowest-level actions. Generally, most of these actions consist of two parts: the robot movement part and the arm (gripper) execution part. For robot movement, we use the A* algorithm to find paths and avoid collisions. We build a connection map during scene initialization for navigation, taking the robot's width into consideration. For the arm (gripper) execution, we primarily use the IK algorithm to compute arm movements. However, since IK cannot handle complex tasks, such as picking objects from the fridge, we also leverage the Open Motion Planning Library (OMPL) planner (Sucan et al., 2012) with forward planning to assist in planning the arm positions.

## B.3 MORE DATASET DETAILS AND DISCUSSION

**Dataset production**    The process of producing data is mainly explained in Section 4.2. In summary, we follow the order of "sample preference - sample scene - sample objects to be manipulated - generate actions guided by a rule-based planner".

**Length and FPS of the simulations**    The length of the simulations ranges from 1 to 5 minutes, depending on the tasks recorded. The videos are recorded at 30 fps.

**Actions contained in each simulation**    The number of actions in simulations varies among different preference levels. There is 1 subtask for action-level, 2-3 subtasks for option-level, and 2-3 subtasks for sequence-level preferences. Each subtask contains 8-20 actions.

**Scenes and rooms**    Each scene contains various types of rooms. The main differences between scenes are the type, number, and layout of both rooms and furniture. Additionally, each room may contain different objects and have unique layouts. Details of the scenes and rooms can be found in Omnigibson's official documentation (https://behavior.stanford.edu/omnigibson/), as we directly adopt these scenes from the open-sourced project.

**290 preference types**    Considering that preferences in household activities are not only multi-dimensional but also hierarchical, we first define a hierarchy of preferences from the perspective of how things happen in a life scenario, that is, from each specific action to a sub-task consisting of several actions, and then to the sequence combining these sub-tasks. The next step is to expand each level with typical tasks and actions. The detailed definition of the 290 preferences can be found in Section 4.1.

**The egocentric view**    Collecting both egocentric observations and third-person views is feasible in PbP or similar environments built on simulators like iGibson. However, in real-world scenarios, it is generally easier to gather egocentric observations of human daily activities, as these can be efficiently captured through wearable devices. Additionally, there are numerous egocentric-view datasets available, such as Ego4D(Grauman et al., 2022), which further facilitate this approach. While third-person views can provide a different perspective, they often encounter issues such as occlusion. Although research based on third-person views is essential for applications involving real robots, focusing on egocentric views in the current work allows for a more straightforward exploration of preference learning and planning. Nevertheless, third-person view data can be obtained by integrating additional cameras, as outlined in our provided code.

**Action ground truth**    In experiments involving vision input, we do not explicitly provide the action sequence of the user. In the symbolic-based experiment, we provide the action sequence to reduce the perception cost to concentrate more effectively on the inference and planning aspects of the study.

Table A2: Action Primitives in PBP.

| Action List | Explanation |
|---|---|
| **Move_to_[]** | Move to a specified location, or a specified room, or a specified object |
| **Rotate_to_[]** | Rotate to a specified orientation or a specified object |
| **Pick_[]** | Pick up an object using the gripper, *e.g.*, "Pick_apple" |
| **Place_[]** | Place an object at a location, *e.g.*, "Place_apple_on_table" |
| **Fill_[]_with_[]** | Fill a container with a substance, *e.g.*, "Fill_glass_with_water" |
| **Pour_[]** | Pour a substance from a container, *e.g.*, "Pour_milk" |
| **Open_[]** | Open an object, *e.g.*, "Open_door" |
| **Close_[]** | Close an object, *e.g.*, "Close_fridge" |
| **Cut_[]** | Cut an object, *e.g.*, "Cut_carrot" |
| **Cook_[]** | Cook an item, *e.g.*, "Cook_pasta" |
| **Wash_[]** | Wash an object, *e.g.*, "Wash_dishes" |
| **Clean_[]** | Clean a surface or object, *e.g.*, "Clean_counter" |
| **Cover_[]** | Cover an object, *e.g.*, "Cover_bowl" |
| **Uncover_[]** | Uncover an object, *e.g.*, "Uncover_bowl" |
| **Toggle_on_[]** | Turn on a device, *e.g.*, "Toggle_on_light" |
| **Toggle_off_[]** | Turn off a device, *e.g.*, "Toggle_off_stove" |
| **Wait_[]** | Wait some time |

## C  EXPERIMENT DETAILS

### C.1  CASE STUDY

We also provide a case with preference *Put fruit on the bed* in the following table Appendix C.1. We present a simplified version of the demonstrations, where all video outputs have been translated into symbol-based action sequences for ease of understanding. Video-based models such as LLaVA-Next and GPT-4V struggle with comprehending preferences and tend to replicate certain action patterns from the video demonstration, such as "move to" and "pick up". Llama3 demonstrates a partial understanding and execution of the preference. It correctly moves to each fruit (grape, banana), picks them up, and places them on the bed. However, it also interacts with the pencil and places it on the bed, which is not required by the preference. Ideally, the pencil should be placed on the table, similar to the pen. On the other hand, GPT-4.1(Symbol) accurately interacts with the grape and banana by moving to each fruit, picking it up, and placing it on the bed. This demonstrates a better understanding and execution of the preference compared to the other models.

## D  BASELINE DETAILS

### D.1  VIVIT

As a baseline, we employ the pure-Transformer-based Video Vision Transformer (ViViT) (Arnab et al., 2021), an end-to-end trainable model with proven capabilities in extracting spatial and temporal information from video inputs. Since it lacks a LLM component, ViViT likely serves as a lower bound for commonsense understanding in PBP tasks.

Inspired by Vision Transformer, ViViT extracts spatio-temporal tokens from the input video and outputs video classification labels for classification. We adopt the ViViT implementation from the official GitHub repo `https://github.com/google-research/scenic`.

Specifically, we utilize a ViViT with an image size of 224 and a patch size of 16. We extract 2 frames per second from the input video and pad them with the last frame. The Transformer architecture with 3 attention heads operates on features of hidden size of 192 and depth of 4. Each attention head operates on a dimension of 64. We train our model for 30 epochs with a learning rate 3e-5. For the few-shot setting, we concatenate the demo videos temporally.

### D.2  LLAVA

Building on more sophisticated architectures, LLaVA (Liu et al., 2024a) represents an end-to-end trainable large multimodal model that integrates vision and text for comprehensive visual-language understanding. We specifically evaluate LLaVA-NeXT, which has been finetuned to excel at zero-shot video understanding tasks.

Following the official implementation of LLaVA from `https://github.com/LLaVA-VL/LLaVA-NeXT`, we test the LLaVA-NeXT-Video-7B-DPO model which is designed for video under-

A5

| Case | ID | Scene | Actions |
|------|-----|-------|---------|
| **Context** | Demo 1 | Beechwood | move to orange, pick up orange, move to bed with orange, place orange on bed, move to apple, pick up apple, move to bed with apple, place apple on bed, move to dvd, pick up dvd, move to table with dvd, place dvd on table. |
| | Demo 2 | Rs | move to banana, pick up banana, move to bed with banana, place banana on bed, move to club sandwich, pick up club sandwich, move to countertop with club sandwich, place club sandwich on countertop, move to orange, pick up orange, move to countertop with orange, move to bed with orange, place orange on bed. |
| | Demo 3 | Merom | move to pen, pick up pen, move to table with pen, place pen, move to strawberry, pick up strawberry, move with to bed with strawberry, place strawberry on bed, move to textbook, pick up textbook, move with textbook, move to bookshelf with textbook, place textbook in bookshelf. |
| **Test Scene** | | Beechwood | grape, pencil, banana |
| **Answer** | LLaVA-Next | Beechwood | move to grape, move to bed with grape, move to pencil, pick up pencil, move to banana, pick up banana |
| | GPT-4V | Beechwood | move to grape, pick up grape, move to bed with grape, place grape on bed |
| | Llama3 | Beechwood | move to grape, pick up grape, move to bed with grape, place grape on bed, move to pencil, pick up pencil, move to bed with pencil, place pencil on bed, move to banana, pick up banana, move to bed with banana, place banana on bed |
| | GPT-4.1(Symbol) | Beechwood | move to grape, pick up grape, move to bed with grape, place grape on bed, move to banana, pick up banana, move to bed with banana, place banana on bed |

Table A3: Case Study with preference *Put fruit on the bed*.

standing. Specifically, we run the model following the default inference settings, with vicuna_v1 as the prompt mode, a sample frame number of 32, and a spatial pooling stride of 2. The textual prompts are as follows[2]:

```
"Stage One / Preference Prediction"
You are a robot assistant that can help summarize the host's preference.
All possible preferences are: {ALL POSSIBLE PREFERENCES}
Now there are some prevous video demos:
[VIDEO_DEMO_1] The preference is [PREFERENCE_1]
[VIDEO_DEMO_2] The preference is [PREFERENCE_2]
[VIDEO_DEMO_3] The preference is [PREFERENCE_3]
Now, please summarize the preference from the last video: [TEST_CASE]
Quesiton: What's the user's preference? Choose from the preference listed
↪   before:

"Stage Two / Planning"
You are a robot assistant. Please view the demos and help generate action
↪   sequence.
All possible preferences are: {ALL POSSIBLE ACTIONS}
Now there are some prevous video demos:
[VIDEO_DEMO_1]
[VIDEO_DEMO_2]
```

---

[2]For the textual prompts, we aim to maintain consistency across all LLMs, although some baselines may have additional requirements for the input format. The prompt design is mainly motivated by OpenAI Cookbook `git@github.com:openai/openai-cookbook.git`. We omitted the prompt tuning process, as we found that minor changes in the prompt were unlikely to significantly impact the results. Conversely, selecting the proper demonstrations in the few-shot examples has a much greater influence on the results.

```
[VIDEO_DEMO_3]
Now you are in the scene with [SCENE DESCRIPTIONS]. Your action sequence
↪  is:
```

### D.3   EILEV

For specialized egocentric video processing, we incorporate EILEV (Yu et al., 2023), which achieves in-context learning through architectural modifications to a pretrained VLM. Our implementation uses OPT-2.7B (Zhang et al., 2022) as the language backbone. The model's pretraining on Ego4D (Grauman et al., 2022) aligns well with PBP's egocentric perspective.

Following the official implementation from `https://github.com/yukw777/EILEV.git`, we test the EILEV model in PBP. There are two reasons we chose EILEV among other VLMs as one of our baselines: 1) EILEV elicits in-context learning through a series of architectural modifications and a unique training process, 2) EILEV is trained using ego-centric data, which is compatible with PBP's input. The textual prompts are as follows. Since EILEV requires the input of the videos and texts to follow a certain pattern for better in-context learning, there are some small modifications to the prompt:

```
"Stage One / Preference Prediction"
You are a robot assistant that can help summarize the host's preference.
All possible preferences are: {ALL POSSIBLE PREFERENCES}
Quesiton: What's the user's preference? Choose from the preference listed
↪  before:
Now there are some prevous video demos:
[VIDEO_DEMO_1] The preference is [PREFERENCE_1]
[VIDEO_DEMO_2] The preference is [PREFERENCE_2]
[VIDEO_DEMO_3] The preference is [PREFERENCE_3]
[TEST_CASE]

"Stage Two / Planning"
You are a robot assistant. Please view the demos and help generate action
↪  sequence.
All possible preferences are: {ALL POSSIBLE ACTIONS}
Now there are some prevous video demos:
[VIDEO_DEMO_1]
[VIDEO_DEMO_2]
[VIDEO_DEMO_3]
Now you are in the scene with [SCENE DESCRIPTIONS]. Your action sequence
↪  is:
```

### D.4   GPT-4V

To benchmark against state-of-the-art visual-language models, we evaluate GPT-4V using the Azure OpenAI API (version "gpt-4-turbo-2024-04-09"). Due to image token limitations, we implement video input subsampling while maintaining temporal coherence. The API has a limit of 10 images per request. Consequently, for the zero-shot setting, we resample each input video to 8 frames of size 224. For the few-shot setting, where we need to input 3 extra video demonstrations, we concatenate 4 images into a frame, thereby obtaining 4 videos in 8 frames, maintaining the same frame number as the previous setting. We test the model with a temperature of 0.05. The textual prompts are as follows:

```
"Stage One / Preference Prediction"
You are a robot assistant that can help summarize the host's preference.
All possible preferences are: {ALL POSSIBLE PREFERENCES}
Now there are some prevous video demos:
[VIDEO_DEMO_1] The preference is [PREFERENCE_1]
[VIDEO_DEMO_2] The preference is [PREFERENCE_2]
[VIDEO_DEMO_3] The preference is [PREFERENCE_3]
Now, please summarize the preference from the last video: [TEST_CASE]
Quesiton: What's the user's preference? Choose from the preference listed
↪  before:

"Stage Two / Planning"
You are a robot assistant. Please view the demos and help generate action
↪  sequence.
All possible preferences are: {ALL POSSIBLE ACTIONS}
```

A7

```
Now there are some prevous video demos:
[VIDEO_DEMO_1]
[VIDEO_DEMO_2]
[VIDEO_DEMO_3]
Now you are in the scene with [SCENE DESCRIPTIONS]. Your action sequence
↪   is:
```

Beyond multimodal approaches, we also evaluate single-modal models that process only action sequences:

## D.5 DAG-OPT

We approach symbolic reasoning by framing the problem as a DAG-Optimization task that uncovers dependency relations between actions and preferences (Zheng et al., 2018). Our implementation uses a score-based NOTEARS model to learn a generalized Structural Equation Model (SEM), following previous few-shot reasoning frameworks (Zhang et al., 2021; Xu et al., 2024b) based on causal dependency structures.

We implement the DAG-Opt baseline following `https://github.com/xunzheng/notears.git`. Specifically, we implement a nonlinear NOTEARS using MLP in evaluation.

To assess pure language understanding, we evaluate advanced LLMs including Llama3 (Touvron et al., 2023) and GPT-4.1 (Achiam et al., 2023) using only action sequence inputs. This approach treats actions as high-level abstractions of egocentric videos, reducing visual complexity while maintaining task semantics. We benchmark Llama3-8B as our baseline against GPT-4.1 as the current state-of-the-art, employing prompt designs informed by the OpenAI Cookbook for optimal few-shot performance.

## D.6 LLAMA3-8B

We test the Llama3 series model with the official scripts from `https://github.com/meta-llama/llama3`. Specifically, we test the 8B instruction-tuned variant "Meta-Llama-3-8B-Instruct" on PBP. We test the model with a temperature of 0.05. The textual prompts are as follows:

```
"Stage One / Preference Prediction"
You are a robot assistant that can help summarize the host's preference.
Please read the following text file and summarize the user's preference.
All possible preferences are: {ALL POSSIBLE PREFERENCES}
[TEXT_ANNOTATION_1] The preference is [PREFERENCE_1]
[TEXT_ANNOTATION_2] The preference is [PREFERENCE_2]
[TEXT_ANNOTATION_3] The preference is [PREFERENCE_3]
Now, please summarize the preference from the last tet file: [TEST_CASE]
Quesiton: What's the user's preference? Choose from the preference listed
↪   before:

"Stage Two / Planning"
You are a robot assistant. Please read the following text files and help
↪   generate action sequence.
All possible preferences are: {ALL POSSIBLE ACTIONS}
Now there are some prevous video demos:
[TEXT_ANNOTATION_1] (action sequence)
[TEXT_ANNOTATION_2] (action sequence)
[TEXT_ANNOTATION_3] (action sequence)
Now you are in the scene with [SCENE DESCRIPTIONS]. Your action sequence
↪   is:
```

## D.7 DEEPSEEK-R1

We use a self-hosted DeepSeek-R1-671B model with a temperature setting of 0.05. For our textual prompts, we deliberately disable the deep thinking capability by inserting an empty thinking section (" $< think >$ \n\n $< /think >$ ") into the assistant content, as our experiments have shown that the deep thinking feature provides no performance improvement for tasks in PBP:

```
"Stage One / Preference Prediction"
You are a robot assistant that can help summarize the host's preference.
Please read the following text file and summarize the user's preference.
All possible preferences are: {ALL POSSIBLE PREFERENCES}
[TEXT_ANNOTATION_1] The preference is [PREFERENCE_1]
[TEXT_ANNOTATION_2] The preference is [PREFERENCE_2]
```

```
[TEXT_ANNOTATION_3] The preference is [PREFERENCE_3]
Now, please summarize the preference from the last tet file: [TEST_CASE]
Quesiton: What's the user's preference? Choose from the preference listed
↪  before:

"Stage Two / Planning"
You are a robot assistant. Please read the following text files and help
↪  generate action sequence.
All possible preferences are: {ALL POSSIBLE ACTIONS}
Now there are some prevous video demos:
[TEXT_ANNOTATION_1] (action sequence)
[TEXT_ANNOTATION_2] (action sequence)
[TEXT_ANNOTATION_3] (action sequence)
Now you are in the scene with [SCENE DESCRIPTIONS]. Your action sequence
↪  is:
<think>\n\n</think>
```

## D.8   GPT-4.1

We use "gpt-4.1-2025-04-14" with a temperature of 0.05. The textual prompts are as follows:

```
"Stage One / Preference Prediction"
You are a robot assistant that can help summarize the host's preference.
Please read the following text file and summarize the user's preference.
All possible preferences are: {ALL POSSIBLE PREFERENCES}
[TEXT_ANNOTATION_1] The preference is [PREFERENCE_1]
[TEXT_ANNOTATION_2] The preference is [PREFERENCE_2]
[TEXT_ANNOTATION_3] The preference is [PREFERENCE_3]
Now, please summarize the preference from the last tet file: [TEST_CASE]
Quesiton: What's the user's preference? Choose from the preference listed
↪  before:

"Stage Two / Planning"
You are a robot assistant. Please read the following text files and help
↪  generate action sequence.
All possible preferences are: {ALL POSSIBLE ACTIONS}
Now there are some prevous video demos:
[TEXT_ANNOTATION_1] (action sequence)
[TEXT_ANNOTATION_2] (action sequence)
[TEXT_ANNOTATION_3] (action sequence)
Now you are in the scene with [SCENE DESCRIPTIONS]. Your action sequence
↪  is:
```

## E   FUTURE DISCUSSION

### E.1   LIMITATIONS AND FUTURE WORK

Our work's primary limitation stems from its reliance on synthetic data. Despite Omniverse's high-quality scene rendering, the simulator cannot fully replicate real-world complexity and variability. Furthermore, human-defined preference labels may not completely capture preference subtleties and diversity. We are addressing these limitations by collecting real-world preference demonstrations using head-worn devices.

