# OpenReview forum: "Learning to Plan with Personalized Preferences"
_ICLR.cc/2026/Conference — ICLR 2026 Conference Withdrawn Submission_

### Official Review · Reviewer_T8qH · 2025-10-30

**Soundness:** 2
**Presentation:** 3
**Contribution:** 2
**Rating:** 4
**Confidence:** 3

**Summary:**

This paper presents a benchmark for testing planning methods' ability to adapt to personalized preferences based on prior demonstrations. For example, the benchmark presents a request of a person asking for food to be prepared when they come home, alongside demonstrations from that person's prior interactions which contain information such as the person's food preferences, eating location preferences, etc., all of which might inform the agent on how to do the task at hand. The result is adaptation to the human that reduces need for clarification questions.

The paper motivates itself with clear examples of preferences and analysis of the distinction between field-standard preferences as in RLHF, language instructions, and here the preferences latent in demonstration information. It goes through significant amounts of cognitive theory, then sets up the benchmark itself: a dataset of action level, option level, and sequence level preferences, plus trajectories relevant to the objects in the preferences, generated synthetically.

Finally, the paper presents experiments using various VLMs and LLM-based ablations. For action-level preferences, it evaluates primarily through the use of Levenshtein distance between the generated trajectory and the ground-truth trajectory. The distances are high, which the paper attributes to the models' failure to understand overall concepts underlying action sequences and instead output individual matching actions. In general, VLMs fail to infer preferences from demonstrations, and while LLMs succeed when preferences are explicitly given, they still fail in end-to-end scenarios where more abstract reasoning is needed.

**Strengths:**

### Quality
- Motivation and background in cognitive literature is useful and more thoughtful than most AI papers in this regard
- Examples are intuitive and interesting
- Preference examples work. While demonstration construction method doesn't seem to imply a natural distribution, I buy that this benchmark is more about specifically testing ability to reason about high-level latent preferences, not necessarily naturalistic preferences. The former is an important step on the way to the later that I agree needs to be tested in itself.
- Experiment setup makes sense - VLMs as main tests and LLMs as ablations do cover the embodied AI aspect of this benchmark.

### Clarity
- Paper is well-written
- First few figures are well-designed

### Originality and significance
This is an important problem that I'm excited to see progress on. I appreciate that this paper attempts to benchmark complex personalization, and I also appreciate the difficulty of even setting this problem. Given that a lot of personalization work is closed-source, product-oriented, and doesn't encourage methods beyond RAG, I consider it original and significant to see a research-oriented benchmark toward this end.

**Weaknesses:**

### Quality
- Though the preference generating method is okay despite not being naturalistic, I don't understand the choice of evaluation metric. Using Levenshtein distance suggests that ground truth should be clearly recoverable, despite these seeming to be fairly open-ended problems with many possible solutions at least at the token level. Examples and justification would help here.
- There are many useful experiments, but the results are presented confusingly - figures with many statistics, followed by general analysis. It would help to have more claims-driven structure with figures and examples that justify each claim; other results can be moved to appendix.
With justification of the evaluation metric and better structure, I think this paper could be significantly improved.

### Clarity
- Tables and data about results are dense and difficult to follow
- The paper provides explanations for poor performance, e.g. poor symbolic reasoning, but those aren't entailed by the results - instead they are some of many possible explanations. Examples that showcase the lack of abstract reasoning, for example, would help build intuition for why the authors feel that this fully characterizes the errors.

### Originality and significance
More attention could be paid to efforts such as reward modeling from video or inverse RL (not standard RLHF-type reward modeling), which attempt to learn preference landscapes from data.

**Questions:**

See above - how do you justify the use of Levenshtein distance for such open-ended problems? Or, are the preferences and demos designed to only have one or a few solutions?

---

### Official Review · Reviewer_BCHT · 2025-11-01

**Soundness:** 3
**Presentation:** 3
**Contribution:** 3
**Rating:** 6
**Confidence:** 3

**Summary:**

This paper introduces the Preference-based Planning (PBP) benchmark, designed to evaluate the ability of embodied AI agents to learn and plan based on individual user preferences. Built on NVIDIA Omniverse and OmniGibson, PBP includes 290 diverse preferences across 50 scenes and three abstraction levels—action, option, and sequence—allowing agents to learn personalized behaviors from few-shot demonstrations. The benchmark highlights that learning preferences as intermediate representations significantly enhances adaptability and planning efficiency. Experimental results show that current multimodal and symbol-based models struggle to infer and generalize user preferences, especially from visual demonstrations.

**Strengths:**

- Proposes a simple yet intuitive framework that enables agents to infer user preferences and generate corresponding actions from few-shot demonstrations.
- Categorizes preferences into a three-tier hierarchical structure (action, option, sequence), capturing different levels of specificity across tasks.

**Weaknesses:**

- The definition of Levenshtein distance is not commonly understood; a clear explanation or brief formal description would help readers unfamiliar with the metric.
- The rule-based and scripted demonstrations lack human variability, causing agents to overfit to artificial planning patterns rather than genuine preference behavior.
- Using Levenshtein distance as the main metric penalizes valid alternative plans and fails to reflect real task success or user satisfaction.
- The comparison is unfair because symbolic models receive ground-truth action inputs, while vision-based models must infer preferences from raw egocentric videos.

**Questions:**

- Would a hybrid approach be possible—for example, performing preference sampling with a rule-based model while delegating action planning to a symbolic model?
- The paper lacks details on how the preference representation model is trained. Could the authors clarify what data and architectures were used for this process?
- During preference prediction, the candidates are explicitly enumerated. Would it be possible to design an open-vocabulary setup where a large language model (LLM) directly infers possible actions and then selects the most preferred one?

---

### Official Review · Reviewer_BwXM · 2025-11-01

**Soundness:** 3
**Presentation:** 3
**Contribution:** 3
**Rating:** 4
**Confidence:** 2

**Summary:**

This paper introduces Preference-Based Planning (PBP), a large simulated benchmark for testing whether AI agents can learn and follow individual human preferences. Built in NVIDIA Omniverse, it includes 50 scenes and 290 preference types, each shown through a few demo videos.

Agents must infer a user’s preference from demonstrations and then plan actions in new scenarios that match that style. The authors compare end-to-end and two-stage approaches, finding that models perform poorly when planning directly from videos but improve dramatically when given or predicting an explicit preference label. Overall, the paper states that PBP shows that learning preferences is the key bottleneck for personalized, human-aware planning in embodied AI.

**Strengths:**

The paper tackles the interesting challenge of teaching embodied AI to learn personalized user preferences. It introduces Preference-Based Planning (PBP), the first large-scale benchmark designed for this purpose. With hundreds of diverse, hierarchical preferences in realistic simulated environments, PBP provides a strong foundation for future research on personalization in planning.

The authors thoroughly test a wide range of models including vision-language systems (ViViT, LLaVA-Next, GPT-4V) to symbolic planners (Llama, DeepSeek), comparing end-to-end and two-stage setups. The results clearly show that explicitly modeling preferences leads to much better plans, confirming the paper’s core hypothesis. This broad, balanced evaluation gives the findings strong credibility.

Beyond accuracy metrics, the analysis explains why two-stage learning works better: it provides a clear preference cue that guides coherent planning, while end-to-end models fail to infer this latent structure. The paper also discusses overfitting issues in vision-only models and the effects of demonstration count, revealing nuanced insights into model behavior.

The writing is good and well-organized, easy to follow, with effective visuals and detailed methodology. The authors commit to open-sourcing the benchmark and code, which has potential reproducibility and long-term community impact.

**Weaknesses:**

One limitation of the work is unclear how well the learned preferences would hold up in messy, real-world conditions where demonstrations are imperfect and preferences don’t fall into neat categories. Even a small-scale user study or robot test discussion would help show that the impact and sim-to-real gap beyond the virtual setup.

The paper also doesn’t introduce a new learning algo, it mostly evaluates existing models like GPT-4, ViViT, and LLaVA within a reasonable two-stage pipeline. The approach makes sense for the problem, but it’s conceptually straightforward, and the novelty lies more in the benchmark and analysis than in new methodology. Some readers might find the contribution more diagnostic than inventive. However, the thorough evaluation across so many baselines gives the work substantial value as a foundation for future research.

In terms of performance, the two-stage design clearly helps, but a big gap remains between using predicted preferences and the true (ground-truth) ones. This shows that current models still struggle to infer user preferences reliably. Moreover, the best results come from massive models like GPT-4.1 and DeepSeek-671B, which are computationally expensive and impractical for deployment. Smaller models such as ViViT or an 8B-parameter Llama perform much worse, raising concerns about scalability and accessibility. It would have been helpful to see a discussion or experiment on how to make preference learning more efficient for lightweight.

While PBP covers an impressive range of preference types, they’re all fixed and labeled. Real preferences are often fuzzier, context-dependent, evolving, and sometimes contradictory. By treating preference inference as a classification problem, the benchmark simplifies the challenge into something more like supervised learning. That’s fine for a first step, but it doesn’t capture the full richness of human personalization. The authors acknowledge this, and it’s worth emphasizing: future work will need to move beyond pre-enumerated labels toward modeling preferences that emerge, shift, and combine dynamically.

**Questions:**

It would be interesting to know how well the approach generalizes beyond the 290 predefined preferences in PBP. In real life, people often have unique habits or styles that weren’t anticipated in the benchmark. Could the agent compose known primitives to infer new ones, or would it need retraining from scratch? Some discussion on how open-ended the preference understanding could be would add depth.

Another open question is real-world deployment. Has the team tried running this on real video demos or physical robots? For example, could it learn how a user likes their tea made from a few wearable-camera clips? Even a small real-world test would help validate how well models trained in simulation hold up when facing noisy perception and less-structured behavior.

The current setup represents preferences as discrete labels, which works well for evaluation but might limit nuance. A learned continuous embedding, encoding each demonstration into a vector the planner can use, could be more flexible and scalable. If the authors explored this, or have thoughts on whether it might outperform classification-based methods, it would be valuable to include.

Finally, there’s the question of efficiency. The best results come from huge models like GPT-4.1 and DeepSeek, which aren’t practical for most applications. Could smaller or specialized models trained directly on PBP data learn preferences more efficiently? Integrating a preference module into a lighter policy network might be one way forward. Insights here could guide future work toward making preference-based planning more accessible and scalable.

---

### Official Review · Reviewer_KUqS · 2025-11-01

**Soundness:** 2
**Presentation:** 1
**Contribution:** 3
**Rating:** 2
**Confidence:** 3

**Summary:**

The motivation is to let the LLM-based agent handle the preferences of individual users and suggest a response that reflects their preferences.
The main task is to learn preference representation with action observations and use the learned preference representation to perform planning in different situations, illustrated in Figure 1.
Figure 2 shows the organization of the data for the preferences collected at three levels, inspired by hierarchical reinforcement learning.
From the description in the figures, they reflect cooking domains.
Section 4 presents the benchmark, with a test set for evaluation.
The methods under evaluation are few-shot learning from demonstrations, typically three demonstrations. (It is not clear how this was actually done.)

**Strengths:**

A summary of the overall data statistics will give a strength due to the large number of collections.
The three different levels of data categorization are also interesting, and this could be a good dataset for RL.
The paper also evaluated vision/language models.

**Weaknesses:**

Sections 1, 2, and 3 don't clearly show what the task is.
It discusses high-level concepts and relevant literature, but it doesn't explain how the benchmark problems are defined/formulated.
It is not clear whether the benchmark is dedicated to the vision-language model or the vision/language only models.
Figure 4 shows the demonstration example with a sequence of figures. This is still confusing to understand what is the input and what is expected to be evalauted.

**Questions:**

Q1 When this benchmark provides the data for capturing the personalized preference,
how does it handle a particular individual? The dataset is a collection of actions/options/sequences from multiple people/individuals.

Q2 How was the in-context learning with demonstration performed over different models and different modalities?

Q3 Why the proposed approach can handle the personalzied preference?

---

### Note · Authors · 2026-01-06

I have read and agree with the venue's withdrawal policy on behalf of myself and my co-authors.